organic chemistry/green chemistry/synthetic chemistry

mechanochemistry, organophosphorus, dinucleotides

**Author for correspondence:**
M. E. Migaud
e-mail: mmigaud@southalabama.edu

This article has been edited by the Royal Society of Chemistry, including the commissioning, peer review process and editorial aspects up to the point of acceptance.

# Investigations into the synthesis of a nucleotide dimer via mechanochemical phosphoramidite chemistry

## C. Johnston[1], C. Hardacre[2] and M. E. Migaud[1,3]

[1]School of Pharmacy, Queen's University Belfast, 97 Lisburn Road, Northern Ireland, UK
[2]The Mill, Sackville Street Campus, University of Manchester, Manchester, UK
[3]Department of Pharmacology, Mitchell Cancer Institute, 1660 Spring Hill Avenue, Mobile, AL, USA

CJ, 0000-0001-5096-9897; CH, 0000-0001-7256-6765; MEM, 0000-0002-9626-2405

Liquid-assisted mechanochemistry as a versatile approach for the coupling of a nucleoside phosphoramidite with a 5′-OH partially protected nucleoside has been investigated. Noted advantages over reported methods were a simplified reaction protocol, a drastic reduction in the use of toxic solvents, the facilitation of mechanochemical reactions through the improved mixing of solid reagents, and low hydrolytic product formation.

## 1. Introduction

Since the development of phosphoramidites for use in oligonucleotide synthesis in the 1980s and evidence of their advantages in solid-supported chemistry [1,2], these reagents and their use in solid-phase chemistry have been the cornerstone of the industrial production of synthetic oligonucleotides. Classic liquid-phase methods, however, remain widely used in the synthesis of short oligomers [3,4]. When combined with solid-phase synthesis, dinucleotide and trinucleotide blocks facilitate purification of oligomer products [5] and are valuable tools for investigations of random mutagenesis [6–8]. Despite the effective use of solution-based methods for synthesizing short nucleotides, several issues remain. Nucleosides are notoriously poorly soluble in many organic solvents, and polar aprotic, toxic and undesirable solvents such as dichloromethane (DCM), dimethylformamide (DMF) or pyridine must be used [9] or alternative protective group strategies must be developed. In Kumar and Poonian's studies of dinucleotide synthesis [10], the levulinyl protection strategy of the 3′-terminal unit could not be applied to cytidine because of poor solubility of the monomer in acetonitrile (MeCN) and tetrahydrofuran (THF), and an alternative tertbutyldimethylsilyl (TBDMS) strategy had to be implemented instead. Having the choice of protecting

**Figure 1.** Mechanochemical formation of a dinucleosidic phosphite **3** and *H*-phosphonate by-product **4**.

group be dictated by solubility issues is restrictive and is not necessarily met with success, e.g. 3′-TBDMS protection and 2′-3′ silyl migration [5].

Furthermore, the use of phosphoramidites for dinucleotide synthesis in solution often requires the use of Schlenk lines and anhydrous solvents. These precautions are crucial to prevent hydrolysis of the nucleoside phosphoramidites. In some cases, overcoming this hydrolysis necessitates a large excess of nucleosidic reagents. Often, an excess of 2–4 equivalents of expensive, complex synthetic nucleoside phosphoramidites is required to achieve complete reactions [11]. For these reasons, alternative methods for the preparation of short dinucleotides that are simple, less wasteful and overcome the limitations of poor nucleoside solubility and hydrolytic instability are needed.

It has been demonstrated that liquid-assisted mechanochemistry, for which solvents are used in equimolar quantities to other reagents, can facilitate reactions otherwise limited by the poor solubility of reagents [12]. Ball-milling allows for the substitution of toxic solvents for alternative ones such as ethyl acetate (EtOAc) and alcohols [9,13]. It has also enabled reactions involving hydrolytically sensitive chlorophosphoramidites and phosphorodiamidites [14–16]. More recently, the mechanochemical synthesis of dinucleotides was reported by Thorpe *et al.* [17] in which the phosphite triester bonds were generated from phosphoramidites upon activation with 5-(ethylthio)-1*H*-tetrazole under solventless conditions.

Herein, a simple ball-milling procedure for the synthesis of a nucleotide dimer is reported. The simplification of reactions by solvent-assisted grinding allows for a reduction in the molar equivalency of solvent and reactants, even when some of the reagents are poorly soluble in the selected solvent. Reduction in the hydrolysis of phosphoramidites correlates with a reduction in the capacity of water acting as solvent. The ability to grind poorly soluble materials also offers more freedom in the selection of protecting groups. To establish and overcome the most common limiting factors controlling the mechanochemically driven coupling of 5′-OH nucleosides to 3′-nucleoside phosphoramidites, the coupling between N4-benzoyl-5′-*O*-(4,4′-dimethoxytrityl)-2′-deoxycytidine-3′-*O*-[*O*-(2-cyanoethyl)-*N*,*N*′-diisopropyl phosphoramidite (DMT-dC(Bz) phosphoramidite) **1** and N6-benzoyl-3′-(*O*-acetyl)-2′-deoxyadenosine (Ac-dA(Bz)) **2** to form the corresponding dinucleosidic phosphite **3** (figure 1) was examined.

## 2. Material and methods

All reactions requiring anhydrous or inert conditions were carried out under an atmosphere of argon or nitrogen in oven-dried glassware. Reactions stirred magnetically were performed using Teflon-coated

**Figure 2.** NMR assignment of NMI.Tf.

**Figure 3.** NMR assignment of Py.TFA.

stirring bars. Ball-milled reactions were performed in oven-dried Retsch 1.5 ml steel vessels, incorporating a single 5 mm steel ball, on a Retsch Mixer Mill MM400, with masses of reagents being weighed directly into the vessels. Organic solvents were introduced into the steel vessels using oven-dried Hamilton syringes. Solvents were evaporated from round-bottom flasks using a rotary evaporator or under high vacuum (0.5–1 mmHg).

Toluene and pyridine were dried by storage over activated 4 Å molecular sieves for 24–72 h under nitrogen, and used for co-evaporation of DMT-dC(Bz) phosphoramidite and Ac-dA(Bz), respectively. 4,5-dicyanoimidazole (DCI), anhydrous DCM and anhydrous MeCN were purchased from Sigma-Aldrich Chemical Company. DMT-dC(Bz) phosphoramidite was purchased from Carbosynth. Ac-dA(Bz) was purchased from Jena Bioscience. 1-hexyl-3-methylimidazolium bis[(trifluoromethyl) sulfonyl]imide ([C$_6$mim][NTf$_2$]) was synthesized in house, whereas 1-hexyl-3-methylimidazolium tris(pentafluoroethyl)trifluorophosphate ([C$_6$mim][FAP]) was generously donated by Merck KGaA.

Solvents used for extractions and chromatography were of technical grade. Purifications were carried out using an automated Biotage system with SNAP cartridges equilibrated before use. $^1$H, $^{13}$C, $^{31}$P and 2D (H-COSY, HSQC) NMR spectra were recorded on Brüker Advance DPX 400 and Brüker Ascend 600. TMS (0 ppm, $^1$H NMR), CDCl$_3$ (77 ppm, $^{13}$C NMR), CD$_3$CN (1.94 ppm, $^1$H NMR, 1.39 ppm, $^{13}$C NMR) and CD$_3$OD (3.31 ppm, $^1$H NMR, 49.15 ppm, $^{13}$C NMR) were used as internal references. The chemical shifts are reported in ppm (parts per million). High-resolution mass spectrometry (HRMS) was recorded on a VG Quattro Triple Quadropole Mass Spectrometer (ES).

## 2.1. Synthesis of *N*-methylimidazolium triflate (NMI.Tf)

In a two-neck round-bottom flask, *N*-methylimidazole (2.45 g, 29.8 mmol, 1 eq) in DCM (10 ml) was added. The solution was stirred at room temperature for 0.5 h, after which triflic acid was added (4.93 g, 32.8 mmol, 1.1 eq). Diethyl ether (10 ml) was then added to the solution to crystallize the salt and the reaction mixture stirred for a further 0.5 h. The suspension was then filtered, and the solids were washed with diethyl ether. The solids were then collected and further solvent removal under reduced pressure (high vacuum) for 2 h gave 72% (4.99 g, 21.5 mmol) of a white solid (figure 2).

$^1$H NMR (400 MHz, CD$_3$OD) ppm 3.96 (s, 3H, H-2), 7.57 (d, 2H, J = 11.5 Hz, H-3, H-4), 8.87 (s, 1H, H-1); $^{13}$C NMR (100 MHz, CD$_3$OD) ppm 36.3 (C-2), 121.1 (C-4), 121.9 (quartet, CF$_3$, J = 318.5 Hz, C-5) 124.7 (C-3), 137.1 (C-1); $^{19}$F NMR (377 MHz, CD$_3$OD) ppm −80.2; MS (ES) Calc. 380.9650 (M$^-$ + (M$^-$ + M$^+$)), Found 380.9607; NMR data in general agreement with previously published work [16].

## 2.2. Synthesis of pyridinium trifluoroacetate (Py.TFA)

Pyridine (3.95 g, 49.9 mmol, 1 eq) was placed in a two-neck round-bottom flask which had been cooled to 0°C. Water (2 ml) was added, followed by slow addition of trifluoroacetic acid (6.27 g, 55.0 mmol, 1.1 eq) and the reaction mixture was left to stir for 2 h. The mixture was then concentrated under high vacuum at 70°C to yield 82% (7.92 g, 40.9 mmol) of a white solid (figure 3).

$^1$H NMR (400 MHz, CD$_3$OD) ppm 8.03–8.09 (m, 2H, 2 x H-2), 8.60 (tt, 1H, J = 8.0 Hz, 1.5 Hz, H-3), 8.85–8.89 (m, 2H, 2 x H-1); $^{13}$C NMR (100 MHz, CD$_3$OD) ppm 118.3 (quartet, CF$_3$, J = 292.5 Hz, C-5),

128.5, 143.98, 147.1 (Aromatic C, C-1, C-2, C-3), 163.2 (C-4) ; $^{19}$F NMR (377 MHz, CD$_3$OD) ppm -77.0 (s, CF$_3$); MS (ES) +ve Calc. 80.0500 (M$^+$), Found 80.0542, -ve Calc. 112.9850 (M-), Found 112.9853; NMR data in general agreement with previously published work [16].

## 2.3. General procedure for optimization of conditions for mechanochemical synthesis of dinucleosidic phosphites

DMT-dC(Bz) phosphoramidite (126 mg, 0.151 mmol, 1.5 eq), Ac-dA(Bz) (40 mg, 0.1 mmol, 1 eq), DCI (30 mg, 0.252 mmol, 2.5 eq) and, if required, either [C$_6$mim][NTf$_2$] (135 mg, 0.302 mmol, 3 eq) or [C$_6$mim][FAP] (185 mg, 0.302 mmol, 3 eq) were added to a Retsch 1.5 ml steel vessel with a 5 mm diameter steel ball bearing. If DCM or MeCN was required, this was added using a Hamilton syringe with the vessel being closed immediately afterwards. The vessel was then shaken on a Retsch MM400 mixer mill at 25 Hz for 0.5–1 h. Following this, the mixture was taken up in CDCl$_3$, and the reaction assessed by examining peak integration ratios by $^{31}$P NMR.

For certain procedures, the reactants were pre-milled in the presence of solvent before the activator was added. The procedure for this was as follows; DMT-dC(Bz) phosphoramidite (126 mg, 0.151 mmol, 1.5 eq) and Ac-dA(Bz) (40 mg, 0.1 mmol, 1 eq) were added to a Retsch 1.5 ml steel vessel with a 5 mm diameter steel ball bearing. DCM (32 µl, 0.503 mmol, 5 eq) or MeCN (26 µl, 0.503 mmol, 5 eq) was added using a Hamilton syringe with the vessel being closed immediately afterwards. The vessel was then shaken on a Retsch MM400 mixer mill at 25 Hz for 5 min. Following this, the vessel was opened, and pre-weighed activator (either DCI (30 mg, 0.252 mmol, 2.5 eq), NMI.Tf (59 mg, 0.252 mmol, 2.5 eq), or Py.TFA (49 mg, 0.252 mmol, 2.5 eq)) was added along with additional DCM (32 µl, 0.503 mmol, 5 eq) or MeCN (26 µl, 0.503 mmol, 5 eq), which were added using a Hamilton syringe. The vessel was closed immediately after addition of the solvent and shaken for a further 1 h at 25 Hz. Following this, the mixture was taken up in CDCl$_3$ and the reaction assessed by examining peak integration ratios by $^{31}$P NMR.

## 2.4. Mechanochemical synthesis of 5′-$O$-DMT-dC(Bz)-3′-5′-[Ac-dA(Bz)], 5

DMT-dC(Bz) phosphoramidite (126 mg, 0.151 mmol, 1.5 eq), co-evaporated three times with dry toluene and Ac-dA(Bz) (40 mg, 0.1 mmol, 1 eq), co-evaporated three times with dry pyridine, were added to a Retsch 1.5 ml steel vessel with a 5 mm diameter steel ball bearing. Anhydrous MeCN (26 µl, 0.503 mmol, 5 eq) was added to the vessel using a Hamilton syringe, with the vessel being closed immediately afterwards. The vessel was then shaken on a Retsch MM400 mixer mill at 25 Hz for 5 min. The vessel was opened and pre-weighed DCI (30 mg, 0.252 mmol, 2.5 eq) was added to the vessel, along with additional MeCN (26 µl, 0.503 mmol, 5 eq). The vessel was closed immediately after addition of the solvent and shaken for a further 1 h at 25 Hz. Following this, the mixture was taken up in CDCl$_3$ and the reaction assessed by examining peak integration ratios present by $^{31}$P NMR, with peak ratios suggesting near-complete consumption of Ac-dA(Bz). The mixture was diluted with a small volume of MeCN, and oxidized using a mixture of I$_2$ (0.1 mol dm$^{-3}$) in pyridine/H$_2$O/THF (1 : 2 : 10 v/v/v) until the iodine colour persisted. The solution was concentrated under reduced pressure and taken up in EtOAc. The organic phase was washed using aqueous sodium thiosulfate 5% w/v (removal of I$_2$ and I$^-$), aqueous NaHCO$_3$ 10% w/v (removal of DCI), and aqueous CuSO$_4$ (removal of traces of pyridine). Undesired phosphodiester, formed by oxidation of the $H$-phosphonate by-product, was removed in the aqueous washes. The mixture was concentrated, redissolved in a minimum volume of DCM, and purified by silica column chromatography using a gradient of 0–10% methanol (MeOH) in EtOAc. Subsequent evaporation of solvent under reduced pressure afforded the product as a white solid. $^1$H and $^{13}$C spectra were complicated due to the formation of a diastereoisomeric mixture (figure 4).

$^1$H NMR (600 MHz, CD$_3$Cl) ppm 2.13 (s, 3H, H-6), 2.14 (s, 3H, H-6), 2.27–2.37 (m, 2H, H-16), 2.60–2.72 (m, 4H, H-2, H-14), 2.73 (t, 2H, J = 6.0 Hz, H-14), 2.87–2.98 (m, 2H, H-16), 3.00–3.11 (m, 2H, H-2), 3.41–3.51 (m, 4H, H-19), 3.77–3.79 (m, 12H, 4 x H-22), 4.05–4.26 (m, 4H, H-13), 4.29–4.44 (m, 8H, H-4, H-5, H-18), 5.09–5.17 (m, 2H, H-17), 5.49–5.55 (m, 2H, H-3), 6.23–6.29 (m, 2H, H-15), 6.51–6.57 (m, 2H, H-1), 6.83–6.88 (m, 8H, 8 x H-21), 7.20–7.38 (m, 22H, Aromatic H (DMT x 18)), 7.45–7.54 (m, 8H, Aromatic H (Benzoyl)), 7.54–7.64 (m, 4H, Aromatic H (Benzoyl)), 7.86–7.92 (m, 4H, Aromatic H (Benzoyl)), 7.98–8.04 (m, 4H, Aromatic H (Benzoyl)), 8.07–8.13 (m, 2H, H-23), 8.31 (s, 1H, H-7), 8.36 (s, 1H, H-7), 8.80 (s, 1H, H-10), 8.81 (s, 1H, H-10), 9.30 (broad s, 2H, NH); $^{13}$C NMR (151 MHz,

**Figure 4.** NMR assignment of 5'-O-DMT-dC(Bz)-3'-5'-[Ac-dA(Bz)], **5**.

**Table 1.** Water contents of materials before drying.

| compound | % water (w/w) | mole % of water |
| --- | --- | --- |
| DMT-dC(Bz) phosphoramidite | 0.76 | 26.2 |
| Ac-dA(Bz) | 0.96 | 17.6 |
| DCI | 0.11 | 0.70 |
| NMI.Tf | 0.75 | 8.88 |
| Py.TFA | 0.46 | 4.68 |
| anhydrous DCM | 0.005 | 0.02 |
| anhydrous MeCN | 0.0085 | 0.02 |

CD$_3$Cl) ppm 19.5–19.7 (C-14), 21.0 and 21.0 (C-6), 37.0 and 37.0 (C-2), 40.2 (C-16), 55.2–55.3 (2 x C-22), 62.2–62.4 (C-5), 62.4–62.6 (C-19), 67.5 and 67.5 (C-13), 74.0 and 74.0 (C-3), 78.5–78.8 (C-17), 83.2–83.3 and 84.9–85.1 (C-4, C-18), 84.4 and 84.5 (C-1), 86.9 and 87.0 (C-15), 87.2 and 87.2 (C-20), 113.4 (C-21), 116.2 and 116.3 (CN), 123.4 and 123.6 (C-8), 127.2 and 127.2 (Aromatic C (DMT)), 127.6 (Aromatic C (Benzoyl)), 127.9–128.2 (Aromatic (DMT, Benzoyl)), 128.8 and 128.8 (Aromatic C (Benzoyl)), 129.0 and 129.0 (Aromatic C (DMT)), 130.0–130.1 (Aromatic C (DMT)), 132.7 and 132.7 (Aromatic C (Benzoyl)), 133.2 (Aromatic C (Benzoyl)), 133.5 (Aromatic C (Benzoyl)), 134.9–135.0 (Aromatic C), 141.4 and 141.5 (C-7), 143.9 (Aromatic C (DMT)), 144.2 (C-23), 149.7–149.8 (C-9), 151.5–151.6 (C-28), 152.7–152.8 (C-10), 158.7–158.8 (Aromatic C (DMT)), 162.3–162.4 (C-25), 164.7–164.9 (C-11, C-27), 170.4 (C = O), 170.4 (C = O), 171.1 (C = O), 171.3 (C = O); $^{31}$P NMR (243 MHz, CD$_3$Cl) ppm -1.75, -1.62; MS (ES) Calc. 1146.3763 (M + H)$^+$, Found 1146.3634, Calc. 1144.3606 (M - H)$^-$, Found 1144.3438

# 3. Results and discussion

Firstly, the water contents of the starting materials used in this chemistry were measured. Karl–Fischer analysis (table 1) indicated that the main source of water in the reaction is from the nucleoside phosphoramidite **1** and nucleoside **2**. These materials are hygroscopic and were therefore dried by co-evaporation with either toluene or pyridine before use. As one of the goals of the study was to avoid the need for a large excess of reactants, it was decided to use 1.5 eq of nucleoside phosphoramidite **1** relative to the nucleoside **2**. An excess was used to overcome an incomplete consumption of

**Table 2.** Outcome of ball-milling reactions of 1 eq of Ac-dA(Bz) **2** (40 mg) with 1.5 eq of DMT-dC(Bz) phosphoramidite **1** and 2.5 eq of activator. Ball-milling frequency used was 25 Hz. Ionic liquids, NMI.Tf and Py.TFA were dried for 2 h under high vacuum before use. If a pre-milling step was used (entries 8–11), this involved milling **1** and **2** with the stated solvent for 5 min, followed by adding the activator, an additional 5 eq of solvent (total 10 eq) and milling for the stated length of time.

| entry | liquid phase | milling time (h) | activator | pre-milling | **1**[a] | **4**[a] | **3**[a] |
|---|---|---|---|---|---|---|---|
| **1** | — | 0.5 | DCI | — | 83 | 17 | 0 |
| **2** | 3 eq [C$_6$mim][FAP] | 0.5 | DCI | — | 80 | 17 | 3 |
| **3** | 3 eq [C$_6$mim][NTf$_2$] | 0.5 | DCI | — | 52 | 38 | 10 |
| **4** | 5 eq DCM | 0.5 | DCI | — | 24 | 56 | 20 |
| **5** | 5 eq MeCN | 0.5 | DCI | — | 15 | 58 | 27 |
| **6** | 5 eq DCM | 1 | DCI | — | 15 | 45 | 40 |
| **7**[b] | 5 eq MeCN | 1 | DCI | — | 0 | 37 | 43 |
| **8** | 5 eq DCM | 1 | DCI | 5 eq DCM | 7 | 43 | 50 |
| **9** | 5 eq MeCN | 1 | DCI | 5 eq MeCN | 0 | 36 | 64 |
| **10** | 5 eq MeCN | 1 | NMI.Tf | 5 eq MeCN | 0 | 42 | 58 |
| **11** | 5 eq MeCN | 1 | Py.TFA | 5 eq MeCN | 0 | 40 | 60 |

[a]Percentages of **1**, **3** and **4** present were determined by $^{31}$P NMR analysis of the impure product in CDCl$_3$. **1** ppm, 149.1 and 148.9, **3** ppm 139.2 and 138.7, **4** ppm 7 and 6.9. The presence of two peaks by $^{31}$P NMR was due to the presence of diastereoisomers in **1**.

[b]Percentages of **1**, **3**, and **4** do not add up to 100 because of substantial formation of another by-product, proposed to be the pyrophosphite formed by a reaction between **1** and **4**. See electronic supplementary material, figure S2.

Ac-dA(Bz) **2**. The outcomes of the reaction were monitored using $^{31}$P NMR spectroscopy, as peaks for the nucleoside phosphoramidite **1**, dinucleotide product **3**, and *H*-phosphonate **4** are individually quantifiable by this method. With 1.5 eq of nucleoside phosphoramidite being used, the product peak would theoretically account for 67% of all integration values when Ac-dA(Bz) is completely consumed to form the dinucleosidic phosphite, and the remainder is a mixture of phosphoramidite **1** and hydrolysed phosphoramidite, consisting mainly of the *H*-phosphonate **4**.

The first coupling reaction attempted was a solid-state reaction between DMT-dC(Bz) phosphoramidite **1** and Ac-dA(Bz) **2**, in the presence of DCI (table 2, entry 1). After 30 min, an inspection of the vessel revealed finely milled particles, suggesting poor mass transfer and contact between the reactants. This observation was confirmed by $^{31}$P NMR, which indicated mostly starting material **1**, with a small amount of degradation to **4** taking place. [C$_6$mim][FAP] was then considered as a solvent (table 2, entry 2), due to previous experience with this ionic liquid in stabilizing hydrolytically sensitive phosphorous compounds under milling conditions [14,16]. Again, visual inspection indicated poor mass transfer, as solid white particles were found in suspension in the ionic liquid phase. $^{31}$P NMR indicated the formation of a small amount of **3**, while most of the starting material had unreacted.

In an attempt to improve mixing and mass transfer, [C$_6$mim][NTf$_2$] was also examined as the liquid phase (table 2, entry 3) to establish whether the solubility of reactants in the ionic liquid phase altered outcomes. If one, or more, of the reagents are soluble or partially soluble in the ionic liquid, improved mixing and better reaction outcomes could be anticipated. Previous work showed solubilities in the range of 35–45 mmol dm$^{-3}$ for DMT-dC(Bz) phosphoramidite in a range of [NTf$_2$]$^-$-based ionic liquids [18]. Yet, the starting materials were found to be poorly reactive under these conditions, suggesting that solubility may not be sufficient to facilitate the reaction by liquid-assisted grinding.

Due to the poor outcomes observed with solventless milling and with the incorporation of ionic liquids, equimolar quantities of solvents were considered instead to facilitate the reaction through solvent-assisted grinding. It is important to emphasize that this type of reaction shares similarities with, but is distinct from, classic solution-phase chemistry [19]. Volumes of solvents to be used are calculated as molecular equivalents, resulting in a molar ratio to reactants, rather than a concentration. It has been proposed that for solvent-assisted grinding, the liquid phase facilitates the mass transfer of reactants throughout the vessel [20].

Here, the solvents selected for solvent-assisted grinding were DCM and MeCN. While the nucleoside Ac-dA(Bz) **2** is soluble in DCM, it is insoluble in MeCN. The amount of each solvent added initially was five molar equivalents (table 2, entries 4–7). The solvents were supplied anhydrous and showed extremely low water contents (table 1). Since it is not possible to perform a ball-milling reaction under anhydrous conditions using the Retsch Mixer Mill MM400, the water content was recorded, rather than being controlled, in order to understand the outcome and the drivers of the reactions, including the side-reactions.

Inspection of the milling vessel after these reactions revealed a paste-like mixture, indicating good mixing of reagents. However, the reactions that were milled for 0.5 h (table 2, entries 4 and 5) showed high levels of *H*-phosphonate **4**. This observation is thought to be due to the hydrolysis of DMT-dC(Bz) phosphoramidite **1** after the addition of $CDCl_3$ to the impure reaction mixture for analysis by $^{31}$P NMR. Extending the milling time to 1 h (table 2, entries 6 and 7) resulted in higher conversion yields of the starting material to the desired product **3**, with 5 eq of MeCN showing slightly higher conversions. NMR data for entry 7 also presented a cluster of peaks between 130 and 128 ppm, believed to be due to the formation of a pyrophosphite as a result of a reaction between the phosphoramidite **1** and the *H*-phosphonate **4** (electronic supplementary material, figure S2). This shows that hydrolysis of the phosphoramidite **1** not only depletes the quantity of this material available for reaction with **2**, but that the *H*-phosphonate **4** formed has the potential to deplete **1** even further.

The best outcomes, however, were found with the incorporation of a pre-milling step in the reaction (table 2, entries 8 and 9). DMT-dC(Bz) phosphoramidite **1** and Ac-dA(Bz) **2** were milled in the presence of 5 eq of solvent for 5 min, followed by the addition of DCI and an additional 5 eq of solvent, to compensate for any evaporation of solvent during the pre-milling step and subsequent opening of the vessel, before milling for 1 h. The vessel was always closed immediately after addition of the solvent. The improved outcomes observed by $^{31}$P NMR must be due to improved mixing before initiation of the reaction and activation of the phosphoramidite.

A comparison of activators was then undertaken (table 2, entries 10 and 11). NMI.Tf and Py.TFA are useful activators for the phosphitylation of nucleosides under ball-milling conditions [16]. These salts are hygroscopic (table 1) and were thus dried under reduced pressure for 2 h. Despite the additional water content, both NMI.Tf and Py.TFA conditions gave yields comparable to that of the DCI-catalysed coupling (table 2, entry 9). All three catalysts led to the hydrolysis of unreacted phosphoramidite. DMT-dC(Bz) phosphoramidite **1** and Ac-dA(Bz) **2** have high water contents (table 1) as supplied from commercial sources, and it is difficult to ensure dryness. The water content of the reagents is likely to drive the *H*-phosphonate formation.

These studies indicate that the optimal conditions for this particular phosphitylation reaction appear to be pre-milling of 1.5 eq of DMT-dC(Bz) phosphoramidite **1** and 1 eq of Ac-dA(Bz) **2** with 5 eq of MeCN for 5 min, followed by the addition of DCI and another 5 eq of MeCN before milling for 1 h. For comparison, the same reaction was performed using a standard solution-based method under $N_2$, with anhydrous DCM as the solvent. Under these conditions, only 21% of $^{31}$P NMR peak integrals corresponded to the desired product **3** after 1 h, increasing to 23% after 3 h, with the *H*-phosphonate **4** accounting for 77%. The low yields obtained under solvent conditions are likely to be due to the inherent difficulties in ensuring reactants are dry and maintained under anhydrous conditions, further exacerbated by the water in organic solvents, now used in a large molar excess (more than ×1000 molar equivalent).

It must be noted that the hydrolysis of DMT-dC(Bz) phosphoramidite **1** occurred more readily in the conventional solution-based reaction than under mechanochemical conditions, even under non-optimal conditions. The volume of solvent used in these mechanochemical reactions (5 or 10 eq in total) is not enough to fully solubilize the reactants and minimizes contact with molecular water. The opportunity for hydrolysis to occur under mechanochemical conditions is reduced due to the incomplete dissolution of reactants in the mobile MeCN phase. Thus, the reduction in the quantities of organic solvents limits the extent of hydrolysis by reducing the actual amount of water present in the reaction vessel, and by controlling how this water can interact with the phosphoramidite. This has been noted as a property of solvent-assisted mechanochemistry whereby reactants spend less time in the solution-phase under mechanochemical conditions, thus reducing the opportunity for side-reactions to occur [19,21]. Consequently, it is an advantage of mechanochemistry that reactions can be performed without an inert atmosphere or Schlenk line techniques [22]. While the result of the solution-based method in this study could have been improved upon with more rigorous drying methods and employing stricter anhydrous techniques, the mechanochemical method does not require an inert atmosphere, and thus presents a simpler and more user-friendly protocol.

**Figure 5.** Reaction conditions investigating the effect of adding the activator after milling of the mixture was finished. Co-evaporation was not performed for these experiments.

It is important to note that caution is needed when interpreting the results of mechanochemical reactions, as the mechanisms by which such reactions occur can sometimes be disputed. For example, supposed cases of 'dry' solid–solid reactions have been argued to have been examples of liquid- or solvent-assisted reactions [19,23,24]. To demonstrate that the reaction was occurring in the milling vessel, and not being facilitated by trace amounts of solvents that had been used for co-evaporation or by addition of CDCl₃ for NMR analyses, a set of experiments was performed (figure 5) in which Ac-dA(Bz) and DMT-dC(Bz) phosphoramidite were used as supplied commercially, with no co-evaporation performed. For these experiments, equivalents of DMT-dC(Bz) phosphoramidite **1** were increased from 1.5 to 2 to account for additional hydrolysis that may occur, and total molar equivalents of MeCN were increased from 10 to 14 to account for the additional mass in the vessel. Furthermore, to eliminate the possibility that the reaction was occurring upon addition of the deuterated solvent for NMR analysis, one reaction (*a*) was performed in which DCI was added after the pre-milling step, as described in table 2 entries 8–11, and the second reaction (*b*) was performed with DCI being added only after milling was complete after 1 h. The deuterated solvent for these experiments was also changed from CDCl₃ to CD₃CN. DMT-dC(Bz) phosphoramidite **1** and the product **3** are both soluble in CD₃CN, whereas Ac-dA(Bz) **2** is not. This reduced the possibility of a reaction occurring after the deuterated solvent had been added to the reaction mixture.

Upon completion of the reactions in figure 5, (*a*) showed ³¹P NMR peaks for the dinucleosidic product **3** and *H*-phosphonate **4** as expected, integrating for 46.5% and 53.5% respectively. This is within range of the expected 50 : 50 outcome with 2 eq of DMT-dC(Bz) phosphoramidite **1**. On the other hand, reaction (*b*) only showed formation of the *H*-phosphonate **4**, with the starting material **1** and dinucleosidic product **3** being completely absent by ³¹P NMR. These outcomes confirm that the coupling reaction occurs under ball-milling in the presence of MeCN. Furthermore, it indicates that the deuterated solvent facilitates the hydrolysis of **1** in the presence of DCI. The physical appearance of (*a*) and (*b*) alone was indicative of the outcome of the reaction; (*a*) showed formation of a clear paste after 1 h of milling, which quickly solidified when disturbed due to evaporation of trace volumes of MeCN. Reaction (*b*), however, was a suspension of white particles, which were clearly visible when the mixture was resuspended in CD₃CN. The visual absence of these particles in reaction (*a*) suggests good consumption of the nucleoside **2**, as confirmed by the formation of the PO₃ product **3** by ³¹P NMR.

The experiments performed in figure 5 have wider ramifications for mechanochemical results reported throughout the literature. Researchers will sometimes analyse the outcomes of ball-milling reactions by directly dissolving mixtures in deuterated solvents for analysis by NMR, or by dissolving the reaction mixture in solvents before purifying the desired compound by liquid–liquid extraction and column chromatography [17]. Phosphoramidites are highly reactive and there is a possibility that with 'solventless' reaction conditions, additional or even unwanted reactions occur upon addition of solvent to the milling vessel. In light of Bowmaker's arguments that many solventless reactions are cases of solvent-assisted grinding [19], it is prudent to demonstrate that such reactions are not being facilitated by trace amounts of solvents in the starting material, or by addition of solvents to the vessel for the purpose of analysis or purification. Here, we confirmed that the reaction does not occur outside the parameters of the ball-milling vessel. Similar experiments should be performed as a matter of course to confirm solventless mechano-reactions are indeed happening in the absence of solvent.

**Figure 6.** Oxidation of the dinucleosidic phosphite to form the phosphate product **5**.

Performing this reaction in MeCN alone presents another advantage of mechanochemistry, which is in the use of less toxic solvents. Ac-dA(Bz) **2** was found to be poorly soluble in a wide range of organic solvents, only being soluble in DCM, pyridine and DMF. The poor solubility of 3′-O,N-protected nucleosides in MeCN has previously been noted [25]. Other studies have shown the utility of mechanochemistry in performing reactions on nucleosides without the use of toxic solvents [9]. In contrast with the solution-based reaction which was performed in DCM, the mechanochemical reaction allowed the use of only 10 molar equivalents of MeCN, achieving satisfactory yields by $^{31}$P NMR. This presents the possibility of performing reactions using nucleosides or nucleoside phosphoramidites which are poorly soluble in solvents like MeCN. In the studies of dinucleotide synthesis by Kumar and Poonian [10], dimers incorporating 3′-O-levulinyl cytidine could not be synthesized due to poor solubility of this monomer in the preferred solvents, MeCN and THF. This forced the use of an alternative protecting group, which may not have been necessary with mechanochemical conditions such as those presented here.

Finally, for completeness, the dinucleosidic phosphite **3** was oxidized to the phosphate **5** using a solution of iodine in water/THF/pyridine (figure 6). After removal of the solvents, the mixture was dissolved in EtOAc, and washed with aqueous solutions of sodium thiosulfate and NaHCO$_3$. This allowed for the removal of excess I$_2$, I$^-$ and the $H$-phosphonate by-product **4** once oxidized. Isolation of the pure dinucleotide phosphate **5** was achieved following silica gel purification in a yield of 26%.

The sequence of experiments thus presented have highlighted the parameters that should be considered to best harness the benefits of mechanochemistry in the formation of phosphodiester bonds in the context of nucleotide chemistry. By optimizing the parameters that can influence the successful outcomes of such coupling reactions, we have identified the following benefits:

(1) We have been able to reduce the molar equivalents of nucleoside phosphoramidite used to a mere 1.5 to 1 ratio to the nucleoside, while maintaining good yields by $^{31}$P NMR. This is a major improvement compared to solid-phase processes.
(2) The E-factor for conventional oligonucleotide synthesis has been quoted as 4700 kg kg$^{-1}$ [26] and so there is a pressing need to reduce the environmental impact of such processes. We have been able to substantially reduce the volumes of solvent used in the reaction.
(3) The reduction in the need for full solvation of reagents offers opportunity for the use of less toxic and hazardous solvents.
(4) The transient solubility of reactants in the mobile phase minimizes the hydrolytic effect of water on the phosphoramidite.
(5) The potential for successful coupling of poorly soluble 3′-O-protected nucleosides under milling conditions offers an alternative means to unsuccessful solvent-based coupling conditions that often require entirely new alternative synthetic strategies.

## 4. Conclusion

In conclusion, the parameters which influence the success of synthesizing nucleosidic dimers by mechanochemistry using minimal amounts of nucleoside phosphoramidites and solvents have been

ascertained. This work highlights why and how solvent-assisted mechanochemistry is poised to make a valuable contribution to the field of oligonucleotides and enable more cost-effective oligo(deoxy)nucleic acid syntheses.

Ethics. Assessments of the environmental impact as well as the physical and health risks associated with the handling of the materials under the conditions described were conducted prior to undertaking each experiment. These risks were appropriately mitigated through good laboratory practice as well as suitable chemical and physical protection.

Data accessibility. The datasets supporting this article have been uploaded as part of the electronic supplementary material.

Authors' contributions. Project conception and design, manuscript writing and editing: C.H. and M.E.M.; practical implementation, experimental design, methods, investigational synthesis and manuscript first draft: C.J.; data analyses, method design: C.J., C.H. and M.E.M.

Competing interests. We declared we have no competing interests.

Funding. C.J. was supported by the Queen's University Belfast postgraduate program.

Acknowledgements. We wish to thank Prof. Colin McCoy and the late Prof. Ken Seddon for their valuable discussion and support.

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
