## [Peer Review File · Royal Society Open Science]

Review History

RSOS-201703.R0 (Original submission)

Review form: Reviewer 1

Is the manuscript scientifically sound in its present form?

Yes

Are the interpretations and conclusions justified by the results?

Yes

Is the language acceptable?

Yes

Do you have any ethical concerns with this paper?

No

Have you any concerns about statistical analyses in this paper?

No

Recommendation?

Accept with minor revision (please list in comments)

Comments to the Author(s)

Reviewer Blind Comments to Author:

The authors present an interesting manuscript entitled "Investigations into the Synthesis of a Nucleotide Dimer via Mechanochemical Phosphoramidite Chemistry". The simplified reaction protocol, reduction in the use of toxic solvents, the facilitation of mechanochemical reactions through the improved mixing of solid reagents, and low hydrolytic product formation are all important outcomes and definitely of interest to the audience.

The abstract is clear and concise.

The introduction is clear and concise.

The materials and methods section is clear and concise, and should enable other to reproduce the research if they so desire.

The results section is clear and concise.

The discussion section is clear and concise.

The figures and legends are good.

Minor:

Please change "(TBDMS)-strategy" to read "(TBDMS) strategy" - although the editorial staff may correct me on that.

US/UK English "synthesizing" should be "synthesising" - I struggle with this myself as I routinely write/edit for journals using either UK/US English and use them interchangeably in emails.

Abbreviations should be explained where first used "EtOAc" - same true for ionic liquids.

In the materials and methods "4,5-dicyanoimidazole" is referred to - this is normally abbreviated to DCI, so it should read "4,5-dicyanoimidazole (DCI)".

Later in the experimental the authors refer to "1'-carbonyldiimidazole" which should be "1,1'-carbonyldiimidazole" which is abbreviated to as CDI.

I'm guessing that 4,5-dicyanoimidazole (DCI) is probably the correct reagent, so please delete 1'-carbonyldiimidazole/CDI.

Review form: Reviewer 2

Is the manuscript scientifically sound in its present form?

Yes

Are the interpretations and conclusions justified by the results?

Yes

Is the language acceptable?

Yes

Do you have any ethical concerns with this paper?

No

Have you any concerns about statistical analyses in this paper?

No

Recommendation?

Accept with minor revision (please list in comments)

Comments to the Author(s)

The manuscript describes the synthesis of a protected dinucleotide using mechanochemical methods. The interest for mechanochemical methods comes from the need of large amounts therapeutic oligonucleotides. At present therapeutic oligonucleotides are prepared by solid-phase phosphoramidite methods that use large amounts of acetonitrile and other ancillary solutions. Mechanochemical method may offer significantly lower consumption of solvents and less excess of phosphoramidites. Mechanochemical methods have been used in nucleoside and very recently a report in the use of these methods for the synthesis of dinucleotides has been recently published (reference 17 in the manuscript). Although the manuscript is not the first paper describing the synthesis of a dinucleotide using mechanochemical methods, the approach and results are different and the present manuscript is well detailed and interesting. As a major concern, the manuscript describes the preparation of one single dinucleotide in a relatively low scale (40 mg of starting 3'-acetylnucleoside). Also 3'-levulinyl nucleosides are more interesting than 3'-acetylnucleosides because the levulinyl can be easier to remove to prepare the protected phosphoramidite dimer building block. Demonstrating that the protocols can be extrapolated to other nucleotides, or in larger amounts (gram-scale) will be desirable. Anyway the work of optimization described in the manuscript is interesting.

Minor points.

The experimental work of the optimization assays explained in Table 2 describe the relative amount of equivalents but it will be good to include also the absolute scale of synthesis because the results may be different in micromol or milimol or mol-scale as the effect of the ambient water on the yields should be different.

The description of the synthesis of the dimer after oxidation and silica gel purification does not contain the isolated yield. It should be included.

Reference 17 does not contain the page numbers and some of the volume number is wrong. The right citation is Chem Eur J 2020, 26, 8857-8861.

Review form: Reviewer 3

Is the manuscript scientifically sound in its present form?

Yes

Are the interpretations and conclusions justified by the results?

Yes

Is the language acceptable?

Yes

Do you have any ethical concerns with this paper?

Yes

Have you any concerns about statistical analyses in this paper?

No

Recommendation?

Accept with minor revision (please list in comments)

Comments to the Author(s)

Very didactic and instructive manuscript that comes timely to address some of the current needs of improved mass efficiency in the field of oligonucleotide synthesis. Only suggestion for the author would be to report the impact of such approaches from an environmental standpoint, whether quantitatively (it might be too early to claim such savings could be made though) or qualitatively.

Decision letter (RSOS-201703.R0)

This year has been very difficult for everyone, and we want to take the opportunity to thank you for your continued support in 2020.

The Royal Society Open Science editorial office will be closed from the evening of Friday 18 December 2020 until Monday 4 January 2021. We will not be responding during this time. If you have received a deadline within this time period, please contact us as soon as possible to allow us to extend the deadline. If you receive any automated messages during this time asking you to meet a deadline, we offer apologies and invite you to respond after the festive period or during normal working hours.

With our best for a peaceful festive period and New Year, and we look forward to working with you in 2021.

Dear Dr Migaud:

Title: Investigations into the Synthesis of a Nucleotide Dimer via Mechanochemical Phosphoramidite Chemistry
Manuscript ID: RSOS-201703

Thank you for submitting the above manuscript to Royal Society Open Science. On behalf of the Editors and the Royal Society of Chemistry, I am pleased to inform you that your manuscript will be accepted for publication in Royal Society Open Science subject to minor revision in accordance with the referee suggestions. Please find the reviewers' comments at the end of this email.

The reviewers and handling editors have recommended publication, but also suggest some minor revisions to your manuscript. Therefore, I invite you to respond to the comments and revise your manuscript.

Because the schedule for publication is very tight, it is a condition of publication that you submit the revised version of your manuscript before 01-Jan-2021. Please note that the revision deadline will expire at 00.00am on this date. If you do not think you will be able to meet this date please let me know immediately.

When submitting your revised manuscript, you will be able to respond to the comments made by the referees and upload a file "Response to Referees" in "Section 6 - File Upload". You can use this to document any changes you make to the original manuscript. In order to expedite the

processing of the revised manuscript, please be as specific as possible in your response to the referees.

Kind regards,
Dr Laura Smith
Publishing Editor, Journals

On behalf of the Subject Editor Professor Anthony Stace and the Associate Editor Dr Andrew Harned.

RSC Associate Editor:

Comments to the Author:

The reviewers have expressed enthusiasm for this work and the manuscript as a whole. They have, however, offered several relatively minor suggestions that should be addressed by the authors.

RSC Subject Editor:
Comments to the Author:
(There are no comments.)

Reviewer comments to Author:
Reviewer: 1

Comments to the Author(s)
Reviewer Blind Comments to Author:

The authors present an interesting manuscript entitled "Investigations into the Synthesis of a Nucleotide Dimer via Mechanochemical Phosphoramidite Chemistry". The simplified reaction protocol, reduction in the use of toxic solvents, the facilitation of mechanochemical reactions through the improved mixing of solid reagents, and low hydrolytic product formation are all important outcomes and definitely of interest to the audience.

The abstract is clear and concise.
The introduction is clear and concise.
The materials and methods section is clear and concise, and should enable other to reproduce the research if they so desire.
The results section is clear and concise.
The discussion section is clear and concise.
The figures and legends are good.

Minor:

Please change "(TBDMS)-strategy" to read "(TBDMS) strategy" - although the editorial staff may correct me on that.

US/UK English "synthesizing" should be "synthesising" - I struggle with this myself as I routinely write/edit for journals using either UK/US English and use them interchangeably in emails.

Abbreviations should be explained where first used "EtOAc" - same true for ionic liquids.

In the materials and methods "4,5-dicyanoimidazole" is referred to - this is normally abbreviated to DCI, so it should read "4,5-dicyanoimidazole (DCI)".
Later in the experimental the authors refer to "1'-carbonyldiimidazole" which should be "1,1'-carbonyldiimidazole" which is abbreviated to as CDI.
I'm guessing that 4,5-dicyanoimidazole (DCI) is probably the correct reagent, so please delete 1'-carbonyldiimidazole/CDI.

Reviewer: 2

Comments to the Author(s)

The manuscript describes the synthesis of a protected dinucleotide using mechanochemical methods. The interest for mechanochemical methods comes from the need of large amounts therapeutic oligonucleotides. At present therapeutic oligonucleotides are prepared by solid-phase phosphoramidite methods that use large amounts of acetonitrile and other ancillary solutions. Mechanochemical method may offer significantly lower consumption of solvents and less excess of phosphoramidites. Mechanochemical methods have been used in nucleoside and very recently a report in the use of these methods for the synthesis of dinucleotides has been recently published

(reference 17 in the manuscript). Although the manuscript is not the first paper describing the synthesis of a dinucleotide using mechanochemical methods, the approach and results are different and the present manuscript is well detailed and interesting. As a major concern, the manuscript describes the preparation of one single dinucleotide in a relatively low scale (40 mg of starting 3'-acetylnucleoside). Also 3'-levulinyl nucleosides are more interesting than 3'-acetylnucleosides because the levulinyl can be easier to remove to prepare the protected phosphoramidite dimer building block. Demonstrating that the protocols can be extrapolated to other nucleotides, or in larger amounts (gram-scale) will be desirable. Anyway the work of optimization described in the manuscript is interesting.

Minor points.

The experimental work of the optimization assays explained in Table 2 describe the relative amount of equivalents but it will be good to include also the absolute scale of synthesis because the results may be different in micromol or milimol or mol-scale as the effect of the ambient water on the yields should be different.

The description of the synthesis of the dimer after oxidation and silica gel purification does not contain the isolated yield. It should be included.

Reference 17 does not contain the page numbers and some of the volume number is wrong. The right citation is Chem Eur J 2020, 26, 8857-8861.

Reviewer: 3

Comments to the Author(s)

Very didactic and instructive manuscript that comes timely to address some of the current needs of improved mass efficiency in the field of oligonucleotide synthesis. Only suggestion for the author would be to report the impact of such approaches from an environmental standpoint, whether quantitatively (it might be too early to claim such savings could be made though) or qualitatively.

Author's Response to Decision Letter for (RSOS-201703.R0)

See Appendix A.

Decision letter (RSOS-201703.R1)

Dear Dr Migaud:

Title: Investigations into the Synthesis of a Nucleotide Dimer via Mechanochemical Phosphoramidite Chemistry
Manuscript ID: RSOS-201703.R1

It is a pleasure to accept your manuscript in its current form for publication in Royal Society Open Science. The chemistry content of Royal Society Open Science is published in collaboration with the Royal Society of Chemistry.

On behalf of the Subject Editor Professor Anthony Stace and the Associate Editor Dr Andrew Harned.

RSC Associate Editor
Comments to the Author:
The authors have satisfactorily addressed all of the concerns raised in the previous review. I believe the current manuscript is now suitable for publication.

Reviewer(s)' Comments to Author:

Appendix A

Investigations into the Synthesis of a Nucleotide Dimer via Mechanochemical Phosphoramidite Chemistry

C. Johnston, C. Hardacre, M. E. Migaud

Response to Referees

Reviewer comments to Author:

Reviewer: 1

Comments to the Author(s)

Reviewer Blind Comments to Author:

The authors present an interesting manuscript entitled "Investigations into the Synthesis of a Nucleotide Dimer via Mechanochemical Phosphoramidite Chemistry". The simplified reaction protocol, reduction in the use of toxic solvents, the facilitation of mechanochemical reactions through the improved mixing of solid reagents, and low hydrolytic product formation are all important outcomes and definitely of interest to the audience.

The abstract is clear and concise.

The introduction is clear and concise.

The materials and methods section is clear and concise, and should enable other to reproduce the research if they so desire.

The results section is clear and concise.

The discussion section is clear and concise.

The figures and legends are good.

Minor:

Please change "(TBDMS)-strategy" to read "(TBDMS) strategy" - although the editorial staff may correct me on that.

US/UK English "synthesizing" should be "synthesising" - I struggle with this myself as I routinely write/edit for journals using either UK/US English and use them interchangeably in emails.

Abbreviations should be explained where first used "EtOAc" - same true for ionic liquids.

In the materials and methods "4,5-dicyanoimidazole" is referred to - this is normally abbreviated to DCI, so it should read "4,5-dicyanoimidazole (DCI)".

Later in the experimental the authors refer to "1'-carbonyldiimidazole" which should be "1,1'-carbonyldiimidazole" which is abbreviated to as CDI.

I'm guessing that 4,5-dicyanoimidazole (DCI) is probably the correct reagent, so please delete 1'-carbonyldiimidazole/CDI.

“(TBDMS)-strategy” has been changed to “(TBDMS) strategy” as suggested.

Language throughout the manuscript has been changed to UK English. "Synthesizing" has been changed to "synthesising" etc.

Care has been taken to ensure consistent application of abbreviations throughout the document. The chemical name has been written in full where it is first used, with the abbreviation used thereafter. This has been done for ethyl acetate (EtOAc) and the ionic liquids.

4,5-dicyanoimidazole is indeed the correct reagent and 1'-carbonyldiimidazole has been deleted accordingly. It has been replaced with "DCI."

Reviewer: 2

Comments to the Author(s)

The manuscript describes the synthesis of a protected dinucleotide using mechanochemical methods. The interest for mechanochemical methods comes from the need of large amounts therapeutic oligonucleotides. At present therapeutic oligonucleotides are prepared by solid-phase phosphoramidite methods that use large amounts of acetonitrile and other ancillary solutions. Mechanochemical method may offer significantly lower consumption of solvents and less excess of phosphoramidites. Mechanochemical methods have been used in nucleoside and very recently a report in the use of these methods for the synthesis of dinucleotides has been recently published (reference 17 in the manuscript). Although the manuscript is not the first paper describing the synthesis of a dinucleotide using mechanochemical methods, the approach and results are different and the present manuscript is well detailed and interesting. As a major concern, the manuscript describes the preparation of one single dinucleotide in a relatively low scale (40 mg of starting 3'-acetylnucleoside). Also 3'-levulinyl nucleosides are more interesting than 3'-acetylnucleosides because the levulinyl can be easier to remove to prepare the protected phosphoramidite dimer building block. Demonstrating that the protocols can be extrapolated to other nucleotides, or in larger amounts (gram-scale) will be desirable. Anyway the work of optimization described in the manuscript is interesting.

Minor points.

The experimental work of the optimization assays explained in Table 2 describe the relative amount of equivalents but it will be good to include also the absolute scale of synthesis because the results may be different in micromol or milimol or mol-scale as the effect of the ambient water on the yields should be different.

The description of the synthesis of the dimer after oxidation and silica gel purification does not contain the isolated yield. It should be included.

Reference 17 does not contain the page numbers and some of the volume number is wrong. The right citation is Chem Eur J 2020, 26, 8857-8861.

Table 2 now indicates 40 mg of Ac-dA(Bz) was used in the caption, so the scale of the reaction is clear to the reader.

The isolated yield has been added for the isolated and purified dimer.

Reference 17 has been corrected and care has been taken to ensure all of the references contain the correct information. The references section did contain some typographical errors, but I could not find further errors with the volume numbers.

Reviewer: 3

Comments to the Author(s)

Very didactic and instructive manuscript that comes timely to address some of the current needs of improved mass efficiency in the field of oligonucleotide synthesis. Only suggestion for the author would be to report the impact of such approaches from an environmental standpoint, whether quantitatively (it might be too early to claim such savings could be made though) or qualitatively.

Reducing the environmental impact of synthetic processes for oligonucleotide synthesis is indeed extremely important. The following sentence has been added to point (2) at the end of the Discussion section: "The E-factor for conventional oligonucleotide synthesis has been quoted as 4700 kg/kg²⁶ and so there is a pressing need to reduce the environmental impact of such processes." Reference 26 has been added accordingly. It should now be clear to the reader that mechanochemical processes, which use solvents as molar equivalents, have a clear role to play with regards to reducing solvent waste.